# Earthquake Hazard Mitigation for Uncertain Building Systems Based on Adaptive Synergetic Control

Ayad Q. Al-Dujaili [1,*], Amjad J. Humaidi [2], Ziyad T. Allawi [3] and Musaab E. Sadiq [4]

1 Electrical Engineering Technical College, Middle Technical University, Baghdad 10022, Iraq
2 Control and Systems Engineering Department, University of Technology, Baghdad 10066, Iraq
3 Department of Computer Engineering, College of Engineering, University of Baghdad, Baghdad 10001, Iraq
4 Ministry of Trade, General Company for Grain Processing, Baghdad 10001, Iraq
* Correspondence: ayad.qasim@mtu.edu.iq

**Abstract:** This study presents an adaptive control scheme based on synergetic control theory for suppressing the vibration of building structures due to earthquake. The control key for the proposed controller is based on a magneto-rheological (MR) damper, which supports the building. According to Lyapunov-based stability analysis, an adaptive synergetic control (ASC) strategy was established under variation of the stiffness and viscosity coefficients in the vibrated building. The control and adaptive laws of the ASC were developed to ensure the stability of the controlled structure. The proposed controller addresses the suppression problem of a single-degree-of-freedom (SDOF) building model, and an earthquake control scenario was conducted and simulated on the basis of earthquake acceleration data recorded from the El Centro Imperial Valley Earthquake. The effectiveness of the adaptive synergetic control was verified and assessed via numerical simulation, and a comparison study was conducted between the adaptive and classical versions of synergetic control (SC). The vibration suppression index was used to evaluate both controllers. The numerical simulation showed the capability of the proposed adaptive controller to stabilize and to suppress the vibration of a building subjected to earthquake. In addition, the adaptive controller successfully kept the estimated viscosity and stiffness coefficients bounded.

**Keywords:** synergetic control; adaptive synergetic control; structural building; earthquake; vibration suppression; stability analysis

## 1. Introduction

An earthquake is a type of natural disaster that leads to serious damage and devastation of infrastructure and buildings due to the accompanying high level of acceleration. According to formal reports, these destructive natural hazards lead to the deaths of 10,000 people every year. For example, the EL-Asnam earthquake occurred in Algeria on 10 October 1980, with a recorded magnitude of 7.2. This earthquake caused the destruction of at least 25,000 housing units and made about 300,000 people homeless, while about 2500 people were killed in the main shock [1]. As such, the security of structures against earthquake movements has become one of the most important issues in the life cycle of a building over the past few decades. Consequently, the design of control techniques that are devoted to the suppression of building vibration due to earthquakes has attracted the interest of many engineering researchers on both theoretical and experimental bases [1].

Passive control systems are made up of components that are embedded in or attached to the framework of buildings, without the use of an external power source in their operation. However, limited control actions may be incorporated in their performance. The design of passive control systems is devoted to enhancing the performance of controlled structures by adjusting the damping and stiffness of quaked buildings. Active control systems, on the other hand, require a large external power input, resulting in high reliability

and efficiency with a tradeoff of high cost. The semi-active control systems are a class of active control systems that are supplied with limited external power sources to monitor the actual motions of building structures and to provide the necessary forces that ensure the stability of the controlled building structures.

Due to their many benefits, magneto-rheological (MR) dampers have recently been studied, along with other semi-active devices, for the purpose of reducing earthquake risk. A magneto-rheological damper is a smart device that is supplied by synthetic fluids. The viscosity of the damper fluid can be adjusted from a liquid state to a semi-solid state in milliseconds by applying an electric current to embedded coils. Mechanical simplicity, strong dynamic range, low power requirements, low cost, and extremely quick and excellent control effort responses are some of the distinguishing features of these dampers [2].

To mitigate structural vibration, the control strategies that incorporate MR dampers in their actions can be classified into two categories. The first type is based on Lyapunov stability analysis, which, in turn, requires a mathematical model of the device [3], while the second type applies intelligent control techniques to develop the control signals, where a mathematical model of the device is not required [4]. The hybrid category is the third type of control technique, mixing both classes of control approaches [5].

Many previous studies have presented various control strategies to suppress the vibration of buildings subjected to earthquakes. The following is a brief review of various control approaches that have been applied for the vibration control of buildings under earthquakes.

In [1], Zizouni et al. proposed regulation control to suppress structural vibration during earthquakes based on a semi-active MR damper. The control design presented a robust feedback law based on a Linear Quadratic Regulator (LQR) to actuate the MR damper with the required control energy to reduce the effect of vibration for a three-story building structure supported by an MR damper on the building's first floor.

In [2], Saban Ç. et al. developed an adaptive control method to suppress the vibration of building structures against earthquake excitation and wind loading by utilizing a semi-active MR damper. Adaptive laws were developed to estimate the unknown parameters of the MR damper, while a nonlinear observer was applied to estimate the unmeasurable internal state variable. The adaptive control law was responsible for generating the necessary input control voltage to energize the MR damper.

In [6], Sayed et al. designed a second-order sliding mode controller to mitigate the vibration of smart structures due to dynamic loadings and earthquakes. An MR fluid damper was the vital element in the control scheme.

In [7], Khaled proposed an intelligent controller based on a neural network (NN) to suppress the effect of vibrations on a three-story scaled structure. The MR damper, which was fixed on the first floor of the building, was actuated on the basis of a linear quadratic (LQ) controller. The control voltage signal applied to the MR damper was generated by a combination of a clipped optimal algorithm with the proposed controller.

In [8], Rahmi used hybrid control schemes, based on a proportional–integral–derivative (PID) controller and chattering-free robust sliding mode controller (SMC), to actuate an active seismic control device attached to a multi-degree-of-freedom structural mechanism to reduce the effect of earthquake vibrations.

In [9], Arash proposed a gain-scheduling fuzzy controller (GSFC) for active structural systems subjected to seismic uncertainties. The parameters of the GSFC were tuned online using the principle of "reinforcement learning". The time-delay problem was solved by integrating the proposed controller with a dynamic state predictor.

In [10], Pei-Ching presented a modern control strategy based on transfer function analysis for controlling a shaking table, which simulates a high-rise building structure under earthquake vibration. The validity and feasibility of the proposed control strategy were verified by synthesizing and combining three linear and nonlinear controllers.

In [11], Borzoo designed nonlinear controllers based on sliding mode theory for a nonlinear-modelled building subjected to earthquake vibrations. The study showed that

the floor displacement of the building structure was significantly reduced under earthquake vibrations.

In [12], Schlacher et al. presented hybrid control for a high-rise and active-damper-supported building structure experiencing earthquakes. The disturbance decoupling problem was solved by approximation and by designing simple and new control laws. Lyapunov-based analysis was conducted to develop and prove the stability of the robust controller to reject variation in a controlled mechanical building.

In [13], Seongkyu and Deokyong presented an intelligent control design based on a neural network to suppress earthquake vibrations in a nonlinear three-story building structure using an active mass damper. This study showed better performance in reducing the response of the building structure and modal energy as compared to the multilayer-perceptron controller and non-controlled system.

In [14], Jiazeng et al. presented a model reference adaptive control scheme to reduce the vibration of building structures due to earthquakes. According to Lyapunov-based stability analysis, the adaptive control laws were established on the basis of backstepping control theory. The proposed control design was constructed by taking into account optimum parameter selection, actuator saturation, parametric effects due to the switching period, and little information in the nonlinear structure model.

In [15], Hasan and Oguz designed a fuzzy-based sliding mode controller to mitigate the effect of earthquake vibrations shaking an eight-story seismic isolation shear building structure. The effect of chattering, accompanying sliding mode control, can be reduced by applying fuzzy logic. The proposed controller showed robustness characteristics against parametric uncertainties, varying dynamic loads, and modeling inaccuracies.

In [16], Falu et al. presented vibration control of finite-time earthquaked linear structures in the presence of a time delay and input saturation. Stability analysis based on the Lyapunov functional was conducted to prove the finite-time stability of the controlled structure and to solve the problem of sufficient conditions arising from tolerated saturation. The study showed that the controlled system was finite-time stable, and the attenuation of vibration was successively achieved within a specified level of disturbance rejection.

In [17], Luyu et al. proposed a control scheme, based on adaptive model reference sliding model control, to suppress the vibration of a multi-degree-of-freedom (MDOF) nonlinear building structure due to earthquake. The building was supported by an active mass damper (AMD), and the control design incorporated a modified unscented Kalman filter (UKF) to estimate the states and unknown parameters to synthesize the required control forces in an adaptive manner.

In [18], Amjad et al. conducted adaptive control design using adaptive backstepping sliding mode control to mitigate the effect of vibrations in a building structure with uncertain building coefficients due to earthquake action. The study addressed a single-degree-of-freedom building, and the numerical simulation showed that better capability of suppression was obtained with the proposed adaptive scheme as compared to the conventional non-adaptive backstepping sliding mode controller.

In [19], J. Enriquez-Z. et al. presented a comparative study based on two passive control schemes to reduce the effect of resonant vibrations of a three-story building-like structure vibrated by an electromechanical shaker at its base. On the third floor of the structure, the first control scheme was established to actuate a Tuned Mass Damper, while an auto-parametric cantilever beam absorber was controlled by the second control scheme. A finite element model was developed to validate the frequency response, and the controlled building structure was verified via both numerical and experimental results.

In [20], A. Anvari and F. Shabani reduced the effect of buffeting response in tall buildings by designing an LQR control scheme based on an Active Tuned Mass Damper (ATMD). As compared to other control schemes, the proposed control technique based on a flexible damper showed better control objectives.

In [21], Rahmi G. and Hakan Y. presented a control design based on a fuzzy logic controller (FLC) and PID controllers to reject the vibration of an earthquaked building

structure in the presence of nonlinear behavior in the soil structure. In representing the dynamic model of the structural system, the nonlinear behavior of the soil was characterized by nonlinear hysteresis and restoring forces. The variation in mass parameters at each story was addressed by the FL controller.

In [22], Trujillo-Franco L. G. et al. presented an online algebraic identification technique for parameters of a flexible structure based on modal decomposition and frequency transform analysis. The proposed identification technique was used to support two control schemes for a perturbed building-like structure subjected to resonant operational conditions. A pendulum vibration absorber was applied to act as an auto-parametric system, and a tuned mass damper was used in order to mitigate resonance excitation.

It has been shown that the control approach based on sliding mode control (SMC) is the most efficient, as compared to other control techniques, when suppression of building vibration is required. However, the chattering phenomenon accompanying this control approach design degrades its efficacy. As such, this study explored synergetic control (SC) to solve this problem. The concept of synergetic control is based on synergetic theory, which is inspired by state-space theory. This control approach has been applied in the control design of highly complex, nonlinear, coupled, and connected systems. This control concept, based on synergetic theory, permits the trajectories of the controlled system to evolve on an invariant manifold selected by the designer and to obtain the desired performance in the simultaneous presence of parametric and nonparametric uncertainties. The following general procedure can be followed in designing nonlinear systems based on synergetic control [23–28]:

- Forming an extended differential equation system that represents the various operations, such as optimizing, achieving fixed values, suppressing disturbances, observing coordinates, etc.;
- Synthesizing *external* controls that work to reduce the extra degree of freedom (DOF) of the extended dynamic system model relative to the final manifold;
- Synthesizing the *internal* controls, which can establish links or relations among the *internal* coordinates of the system. The control goal can be guaranteed and assured by these links.

In view of synergetic theory, the synergetic controller guides the trajectories of dynamic system trajectories to move, onto the manifold, from any initial states to their equilibrium states.

In this study, synergetic control theory was applied to develop a novel nonlinear control approach to mitigate the vibration in an MR-damper-supported building structure due to earthquake. Two SC-based controllers were designed to solve the control problem in earthquaked buildings. The conventional synergetic controller (CSC) is the first candidate controller, while the second controller was established on the basis of the adaptive synergetic controller (ASC).

In adaptive control techniques, the parameters of a plant in real time are adjusted in order to maintain a desired level of dynamic performance when the system is subjected to unknown and varying (changing with time) parameters [29–33]. The adaptive version of control based on the synergetic methodology is devoted to solving the problem of uncertainties in the construction device's parameters due to the effect of earthquake by actuating the necessary force via the MR damper. The stability of the earthquaked controlled structure was analyzed and proven on the basis of Lyapunov theory. The contributions of the work can be summarized by the following points:

- Developing classical and adaptive synergetic control algorithms to solve the vibration control problem in earthquaked buildings on the basis of MR dampers and Lyapunov stability analysis;
- Proving asymptotic stability for building systems controlled by classical and adaptive synergetic control schemes, such that all errors finally converge to their corresponding zero equilibrium points based on the Lyapunov theorem;

- Guaranteeing the boundedness of the estimated viscosity and stiffness coefficients of building systems;
- Conducting a comparison between the proposed classical and adaptive synergetic controllers in terms of vibration suppression capabilities.

The rest of the article is organized as follows: A dynamic model of a one-story building is developed in Section 2, and the control methodology based on synergetic control to suppress the vibrating action of an earthquake is analyzed for both classical and adaptive schemes in Section 3. In Section 4, the results of a computer simulation conducted to verify the effectiveness of the proposed control scheme are presented. The concluded points based on observations of the simulated results are highlighted in Section 5. This section also includes suggestions for future extension.

## 2. Dynamic Model

Figure 1 shows a one-story building structure model supported by an MR damper. The MR damper is fitted to the ground and a brace mounted on the first floor. Previous studies showed that fitting the MR damper on the first floor leads to more powerful control forces than does fitting it on upper floors [34,35]. Seismically, this structure is susceptible to acceleration $\ddot{x}_g$.

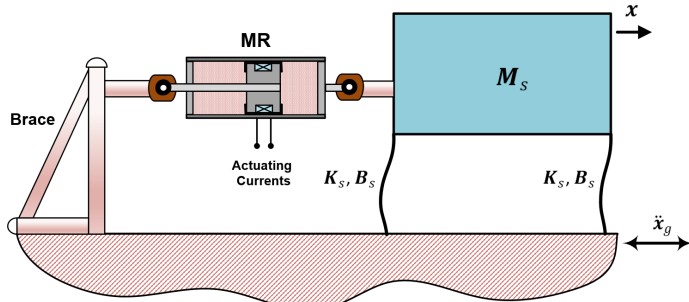

**Figure 1.** One-story building structure supported by an MR damper and vibrated by an earthquake.

The general dynamic motion for the structural system shown in Figure 1 can be described by

$$M_s\,\ddot{x} + B_s\,\dot{x} + K_s\,x = M_s\,\ddot{x}_g + f_{MR} \tag{1}$$

where the variables $x$, $\dot{x}$, and $\ddot{x}$ represent the displacement, velocity, and acceleration of the base floor, respectively. The parameters $K_s$, $C_s$, and $M_s$ represent the stiffness, damping, and mass of the structural system, respectively. The earthquake data function is represented by $\ddot{x}_g$, while the input force generated by the MR damper is indicated by $f_{MR}$. The state variable representation of Equation (1) can be described as follows:

$$\dot{x}_1 = x_2$$

$$\dot{x}_2 = -\frac{B_s}{M_s}\,x_2 - \frac{K_s}{M_s}x_1 + \frac{1}{M_s}v + \ddot{x}_g \tag{2}$$

where the control signal is denoted by $v$ and the earthquake data function is denoted by $\ddot{x}_g$.

Figure 2 shows the constituting elements of the MR damper. As shown in the figure, the conventional damper changes its viscosity according to level of current flowing into the actuating coil. The actuating current is established by the control voltage excited by the controller. The nonlinear model of an MR damper was firstly established and developed by Bingham. Then, other models of MR dampers based on differential equations were developed by other researchers. These updated versions of MR damper models have been validated and verified in experimental tests [36,37].

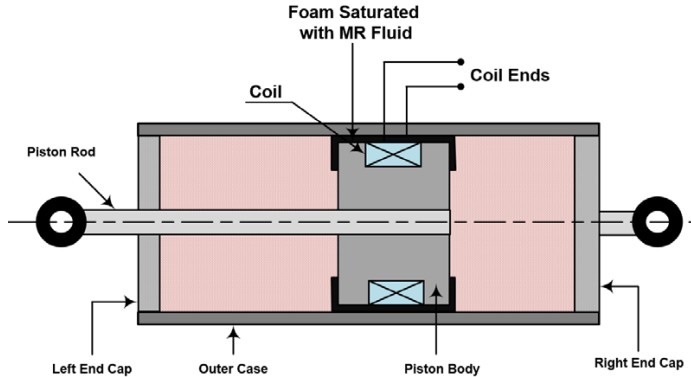

**Figure 2.** Schematic representation of the MR damper.

As shown in Figure 3, the Bouc–Wen model was used to augment the dynamic properties of the MR device. According to Figure 3, the dynamic equations characterizing the MR device can be written as [38]

$$f_{MR} = c_1 \, \dot{y} + k_1 (x - x_o) \tag{3}$$

$$c_1 \, \dot{y} = c_o \left( \dot{x} - \dot{y} \right) + k_o (x - y) + \alpha \, z \tag{4}$$

$$\dot{z} = -\gamma z \left| \dot{x} - \dot{y} \right| |z|^{n-1} - \beta \left( \dot{x} - \dot{y} \right) |z|^n + A \left( \dot{x} - \dot{y} \right) \tag{5}$$

$$\dot{y} = \frac{\left( \alpha \, z + c_o \, \dot{x} + k_o (x - y) \right)}{c_o + c_1} \tag{6}$$

where $f_{MR}$ represents the output force resulting from the MR damper. The linear displacement and velocity of the damper are, respectively, represented by the state variables $x$ and $\dot{x}$. The variable $z$ is an evolutionary variable that represents the hysteretic behavior of the output force in terms of the MR damper's displacement and velocity. The coefficients $k_o$ and $c_o$ denote, respectively, the accumulator stiffness and viscous damping at low velocity coefficients, while the coefficients $c_1$ and $k_1$ represent the damping and stiffness at high velocities. The parameters $\gamma$, $\beta$, $A$, and $n$ characterize the scale and shape of the hysteresis loop. It is worth noting that the above-mentioned parameters concerning the MR damper are related to the applied voltage $v$ [38]:

$$\alpha = \alpha_a + \alpha_b \, u, \; c_o = c_{oa} + c_{ob} \, u, \; c_1 = c_{1a} + c_{1b} \, u \tag{7}$$

$$\dot{u} = -\eta \left( u - v_a \right) \tag{8}$$

where the coefficients $\alpha_a$, $\alpha_b$, $c_{oa}$, $c_{ob}$ $c_{1a}$, and $c_{1a}$ represent the components of damping coefficients $\alpha$, $c_o$, and $c_1$. The parameter $\eta$ is a constant related to the time response, $v_a$ is the control commanded voltage applied to the damper control circuit, and $u$ is the phenomenological variable representing the system dynamics.

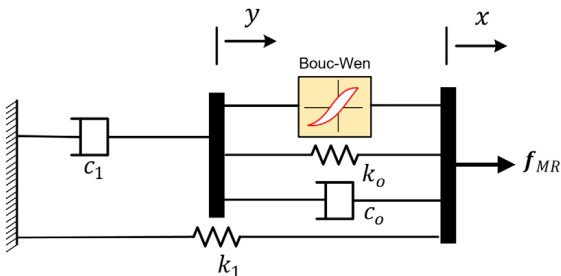

**Figure 3.** Modified Bouc–Wen model of the MR damper.

### 3. Adaptive Synergetic Control Design

In this section, adaptive control design is developed on the basis of a synergetic methodology for suppressing the vibration of structures due to earthquake. Firstly, the classical synergetic control is developed. Then, adaptive synergetic control design is conducted to yield the control and adaptive laws to address the variation in the coefficients of the structural system as a consequence of an earthquake.

*3.1. Design of Classical Synergetic Control*

Let us define the error $e$ to be the difference between the actual state $x_1$ and the desired state (displacement) $x_{1d}$:

$$e = x_1 - x_{1d} \tag{9}$$

In the present application, $x_{1d}$ must be set to zero; then, $e = x_1$. Taking the first and second derivatives of error results in

$$\dot{e} = \dot{x}_1 = x_2$$

$$\ddot{e} = \dot{x}_2 = -\frac{B_s}{M_s}x_2 - \frac{K_s}{M}x_1 + \frac{1}{M_s}v + \ddot{x}_g \tag{10}$$

On the basis of the system dynamic described by Equation (2), one can define the macro-variable $\psi(x)$ as follows:

$$\psi(e) = c \cdot e + \dot{e} \tag{11}$$

The first derivative of Equation (11) is given by

$$\dot{\psi}(e) = c\,\dot{e} + \ddot{e} \tag{12}$$

where $c$ is a scalar design parameter. The macro-dynamic variable's evolution towards the manifolds is defined as follows [23,27]:

$$T \cdot \dot{\psi}(e) + \psi(e) = 0 \tag{13}$$

where $T > 0$ is the convergence rate of the evolved micro-variable of the system. This dynamic is dependent to ensure that the trajectories of all state variables achieve the desired manifold and remain on it for future time [23,27].

On the basis of Equations (9)–(13), one can obtain

$$T\left(c\,\dot{e} + \ddot{e}\right) + \psi\left(e\right) = 0$$

or

$$T\,\dot{x}_2 + T\,c\,\dot{e} + \psi\left(e\right) = 0 \tag{14}$$

Using Equation (2), Equation (14) becomes

$$T\left(-\frac{B_s}{M_s}x_2 - \frac{K_s}{M_s}x_1 + \frac{1}{M_s}v + \ddot{x}_g\right) + Tc\,\dot{e} + \psi(e) = 0 \tag{15}$$

In order to satisfy Equation (15), the control signal is set as follows:

$$v = B_s\,x_2 + K_s\,x_1 - M_s\ddot{x}_g - M_s\,c\,\dot{e} - \frac{M_s}{T}\psi(e) \tag{16}$$

*3.2. Adaptive Synergetic Control*

Due to earthquake, uncertainty in the coefficients of a vibrated building structure will appear. For this study, two uncertainties are considered; one concerns the coefficient of

damping viscosity $B$, and the other uncertainty is due to change in the stiffness coefficient $K$ of the building system [39,40]. This can be expressed as

$$\hat{B} = B + \widetilde{B}, \ \hat{K} = K + \widetilde{K} \tag{17}$$

where $\hat{B}$ and $\hat{K}$ are the estimated values of the nominal viscous damping coefficient $B$ and the nominal stiffness coefficient $K$, respectively. The adaptive synergetic controller is designed on the basis of Lyapunov stability analysis to develop the control and adaptive laws for estimating uncertain coefficients. The objective of the proposed adaptive scheme is to reduce the vibration effect on an MR damper-supported building structure due to earthquake.

**Assumption:** The stability analysis is based on the assumption that the actual values of the stiffness and viscosity coefficients are stationary or vary slowly with time.

The candidate Lyapunov function can be defined as

$$V = \frac{1}{2}\psi(e)^2 + \frac{1}{2}\gamma_1 \widetilde{B}^2 + \frac{1}{2}\gamma_2 \widetilde{K}^2 \tag{18}$$

where $\gamma_1, \gamma_2$ denote the adaptation gains. The time derivative of Equation (18) is given by

$$\dot{V} = \psi(e)\left(c.\dot{e} + \ddot{e}\right) + \gamma_1 \widetilde{B}.\dot{\hat{B}} + \gamma_2 \widetilde{K}.\dot{\hat{K}} \tag{19}$$

Substituting Equation (10) into Equation (19), we obtain

$$\dot{V} = \psi(e)\left(c\,\dot{e} + \left(-\frac{B_s}{M_s}x_2 - \frac{K_s}{M}x_1 + \frac{1}{M_s}v + \ddot{x}_g\right)\right) + \gamma_1 \widetilde{B}\dot{\hat{B}} + \gamma_2 \widetilde{K}\dot{\hat{K}} \tag{20}$$

The control law $v$ can be determined on the basis of Equation (16) using the chosen estimated values of

$$v = \hat{B}_s x_2 + \hat{K}_s x_1 - M_s \ddot{x}_g - M_s c\,\dot{e} - \frac{M_s}{T}\psi(e) \tag{21}$$

Using Equation (21), the expression for $\dot{V}$ becomes

$$\dot{V} = \psi(e)\left(-\frac{B_s}{M_s}x_2 - \frac{K_s}{M_s}x_1 + \frac{\hat{B}_s}{M_s}x_2 + \frac{\hat{K}_s}{M_s}x_1 - \frac{\psi(e)}{T}\right) + \gamma_1 \widetilde{B}\dot{\hat{B}} + \gamma_2 \widetilde{K}\dot{\hat{K}} \tag{22}$$

Using Equation (17), we have

$$\dot{V} = -\psi^2(e) + \frac{\widetilde{B}}{M_s}x_2\,\psi(e) + \frac{\widetilde{K}}{M_s}x_1\,\psi(e) + \gamma_1 \widetilde{B}\dot{\hat{B}} + \gamma_2 \widetilde{K}\dot{\hat{K}} \tag{23}$$

or

$$\dot{V} = -\psi^2(e) + \widetilde{B}\left(\gamma_1\dot{\hat{B}} + \psi(e)\frac{x_2}{M_s}\right) + \widetilde{B}\left(\gamma_2\dot{\hat{K}} + \psi(e)\frac{x_1}{M_s}\right) \tag{24}$$

To guarantee $\dot{V} < 0$, the last two terms must be evaluated as zeros; that is,

$$\gamma_1\dot{\hat{B}} + \psi(e)\frac{x_2}{M_s} = 0 \tag{25}$$

$$\gamma_2\dot{\hat{K}} + \psi(e)\frac{x_1}{M_s} = 0 \tag{26}$$

This produces the adaptive synergetic laws of an adaptive-controlled structure subjected to earthquake:

$$\dot{\hat{B}} = -\frac{x_2}{\gamma_1 M_s}\psi(e) \tag{27}$$

$$\dot{K} = -\frac{x_1}{\gamma_2 \, M_s} \psi(e) \tag{28}$$

On the basis of the adaptive laws given by Equations (27) and (28), $\dot{V}$ is negative definite, and this ensures the asymptotic stability of the earthquaked one-story structure assisted by the MR damper and controlled by the adaptive synergetic controller.

### 4. Computer Simulation

In this section, the effectiveness of the classical and adaptive synergetic controllers is verified; a comparison study of their performance was conducted to show which controller could perform better in terms of vibration suppression during an earthquake. The numerical simulations were conducted within the MATLAB/SIMULINK environment. The "ODE45" numerical solver with variable-step time was chosen. The variable hit ranged between min-step-size $10^{-4}$ and max-step-size $10^{-3}$. The Simulink modeling of the modified Bouc–Wen model is shown in Figure 4. As shown in the figure, this Simulink model was utilized to solve the equation of the z-variable. All constituting elements of the model solver of the z-variable were supplied by the SIMULINK library.

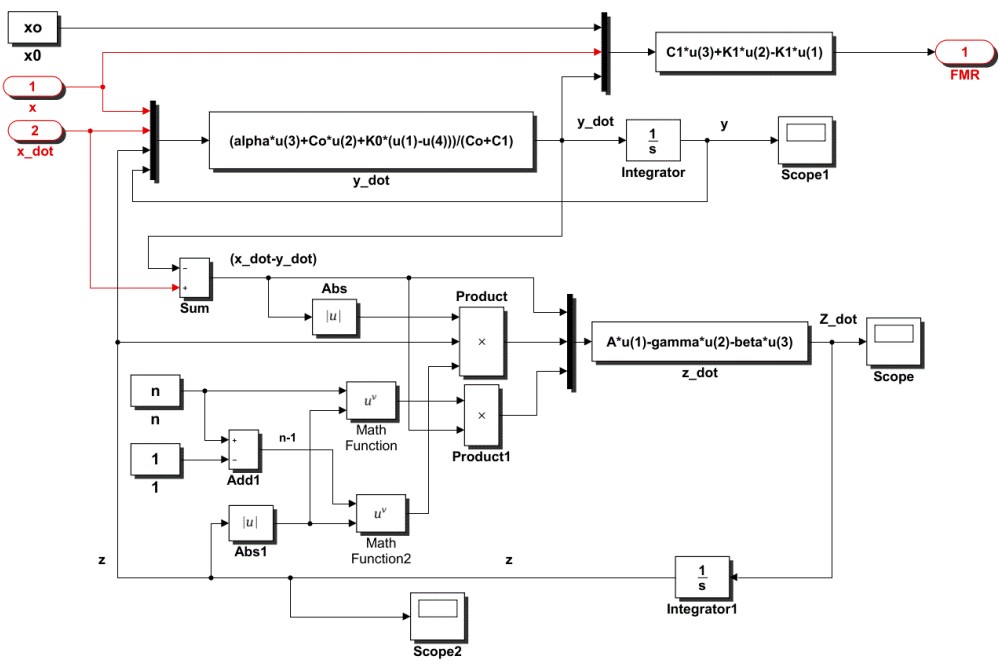

**Figure 4.** The Simulink model of modified Bouc–Wen model.

Table 1 lists the numerical values of the MR damper's parameters. In the case of conventional SC, the design parameters of the controller were chosen to be $T = 1$, $c = 10$. In the case of adaptive SC, the selected design parameters of the controller were $c = 10$, $T = 1$. Moreover, the parameters of the adaptive rates were set as follows: $\gamma_1 = \gamma_1 = 1$. On the other hand, the parameters $M_s = 93.3$ Kg, $C_s = 175$ N·s/m, and $K_s = 12 \times 10^5$ N/m were used for the single-degree-of-freedom building structural system.

In this study, the earthquake data applied for numerical simulations were based on an earthquake disaster that occurred at Imperial Valley (Brawley) in May 1940. The magnitude of the earthquake was estimated at approximately 7.1 on the Richter scale. Figure 5 displays the behavior of the recorded earthquake. Figure 6 depicts the dynamic behaviors of building displacements due to earthquake vibration under the classical synergetic controller and adaptive synergetic controller. According to Figure 6, both the conventional and adaptive control approaches could successfully suppress the effect of earthquake shaking as compared to an uncontrolled system with the support of an MR damper. On the

other hand, the adaptive synergetic controller had better rejection capability in terms of vibration suppression as compared to the classical synergetic controller.

**Table 1.** The numerical values of the MR damper's parameters.

| Simulation Parameter | Value |
|---|---|
| $c_{0a}$ | 21 N·s/cm |
| $c_{1a}$ | 283 N·s/cm |
| $c_{0b}$ | 3.5 N·s/cm |
| $c_{1b}$ | 2.95 N·s/cm |
| $\gamma$ | 363 cm$^{-2}$ |
| $n$ | 2 |
| $x_o$ | 14.3 cm |
| $\beta$ | 363 cm$^{-2}$ |
| $A$ | 301 |
| $k_o$ | 46.9 N/cm |
| $k_1$ | 5 N/cm |
| $\eta$ | 190 s$^{-1}$ |
| $\alpha_a$ | 140 N/cm |
| $\alpha_b$ | 695 N/cm |

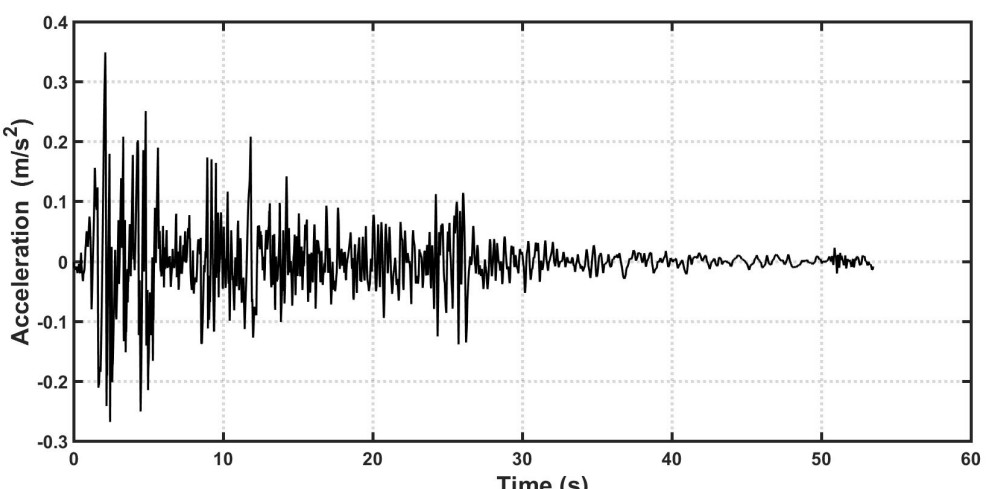

**Figure 5.** El Centro 1940 Imperial Valley Earthquake.

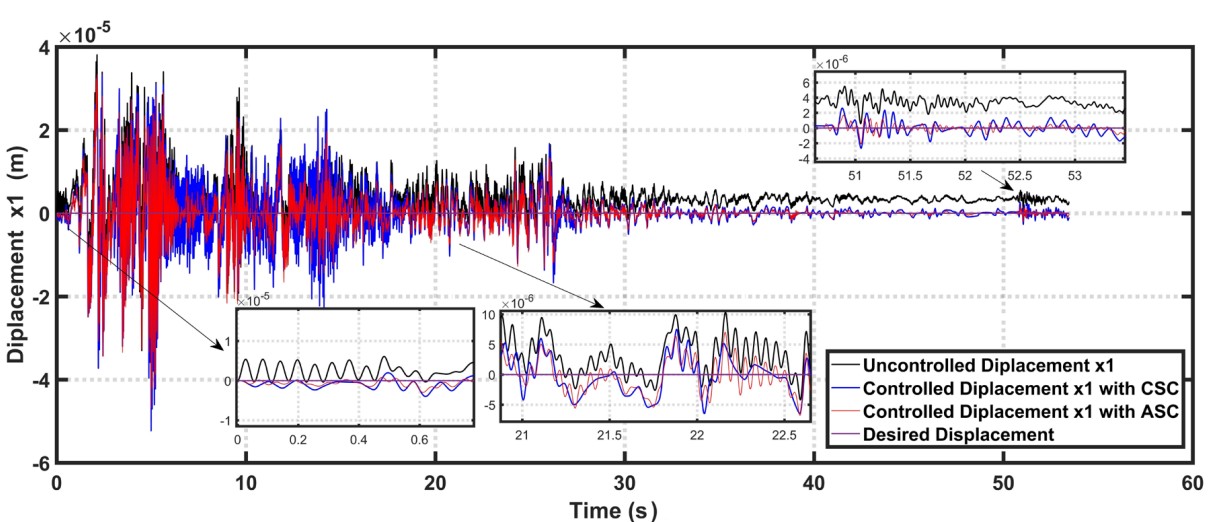

**Figure 6.** Displacement responses of the earth-quaked building based on SC and ASC.

The Root Mean Square (RMS) was chosen as the index of evaluation, and a comparison between SMC and ABSMC was made in terms of displacement deviation. In the case of an uncontrolled building structure, the RMS value in displacement deviation over the time of the earthquake was evaluated to be $13.21 \times 10^{-5}$, while the conventional SC recorded an RMS value of $4.23 \times 10^{-4}$. The lowest value of displacement RMS was calculated when involving adaptive SC over the entire time of the earthquake, at $1.017 \times 10^{-3}$. As such, adaptive SC showed better rejection capability to suppress the level of vibration on the building structure due to the earthquake.

Figure 7 shows the acceleration generated by the adaptive synergetic controller and the conventional synergetic controller. According to Figure 7, one can notice that the adaptive synergetic controller resulted in a lower acceleration value in the quaked building, as compared to the larger acceleration due to the classical SC. Thus, the adaptive synergetic controller is better able to suppress building vibration due to earthquake as compared to the conventional synergetic controller.

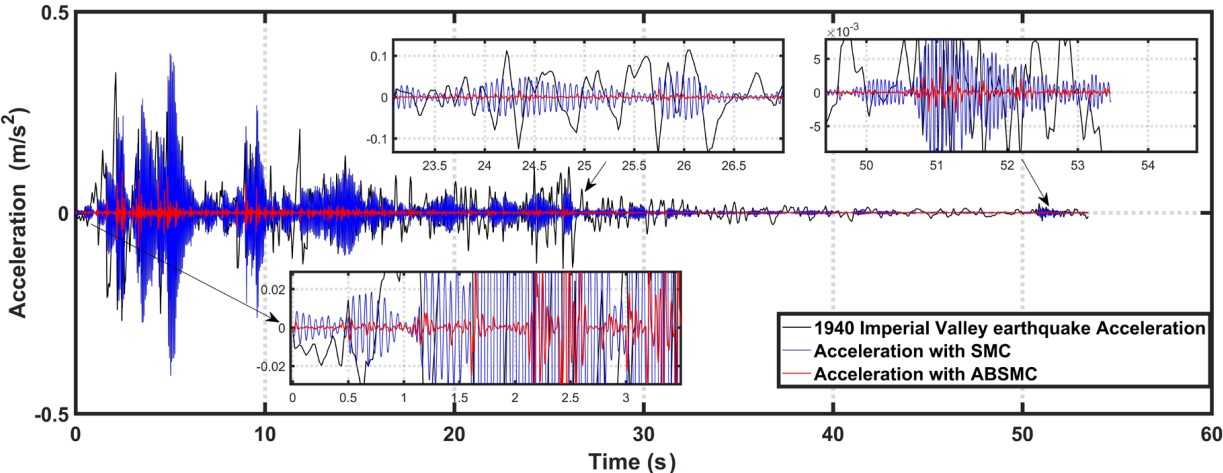

**Figure 7.** Acceleration responses of the earthquaked building based on SC and ASC.

The control signals generated by the classical and adaptive synergetic controllers are indicated in Figure 8. The figure indicates that the control effort in the case of classical SC was less than the control effort produced by ASC. Of course, this the price paid by ASC for its better capability in terms of vibration suppression in the building structure.

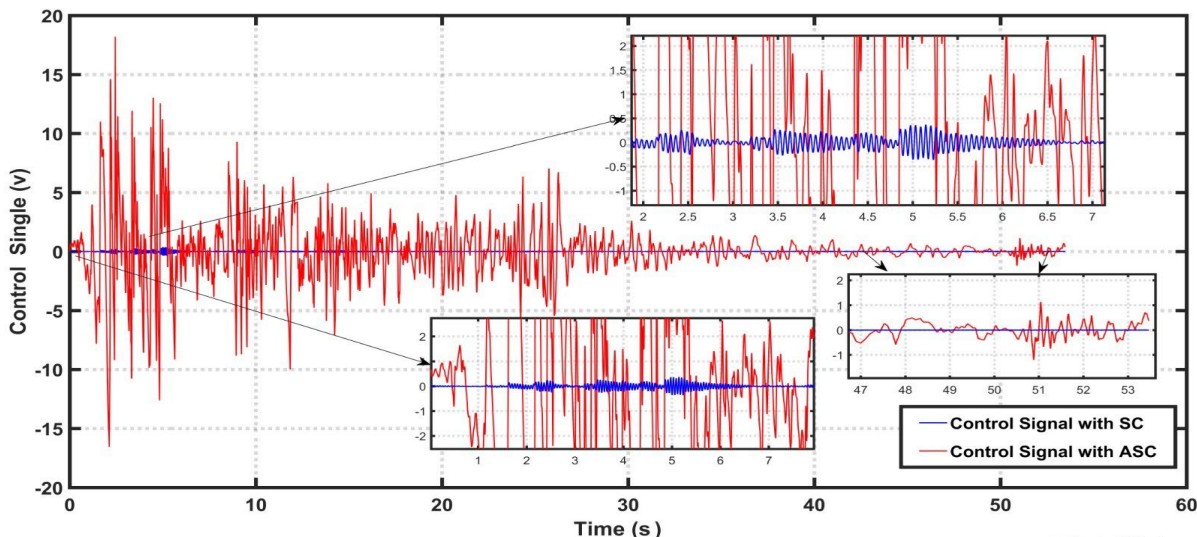

**Figure 8.** Control signals generated by the SC and ASC.

Figures 9 and 10 show the behaviors of uncertainty estimates, generated by the adaptive laws, for both coefficients of building damping viscosity and stiffness, respectively. It is evident from the figures that the estimates of the coefficients produced by the adaptive laws led to bounded estimates of uncertainties, which, in turn, led to stable controlled building systems. This boundedness of estimates could guarantee the avoidance of instability problems due to unbounded estimates. Otherwise, the estimated coefficients could grow without bounds, leading to instability problems in the adaptive controlled building system.

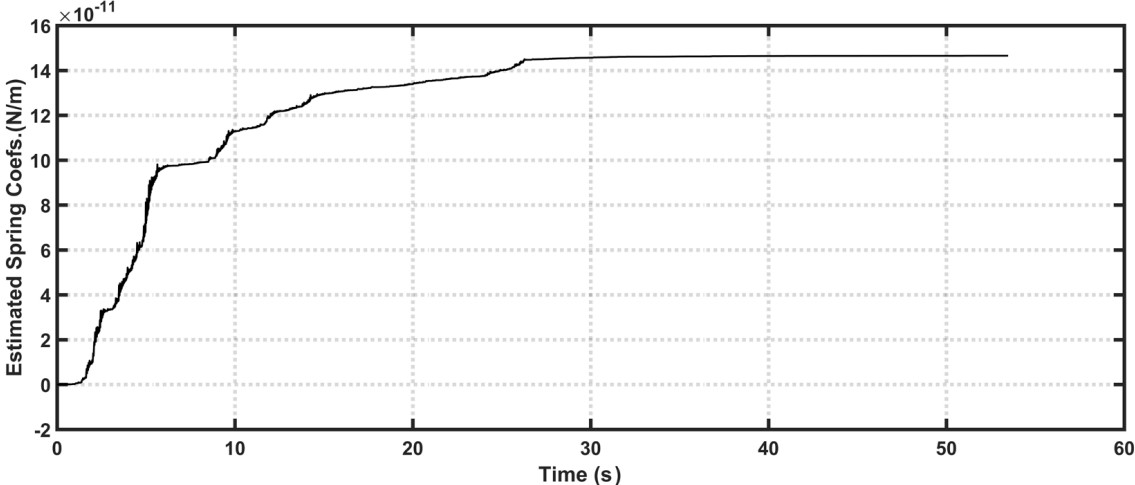

**Figure 9.** Estimated value of stiffness coefficient by adaptive law.

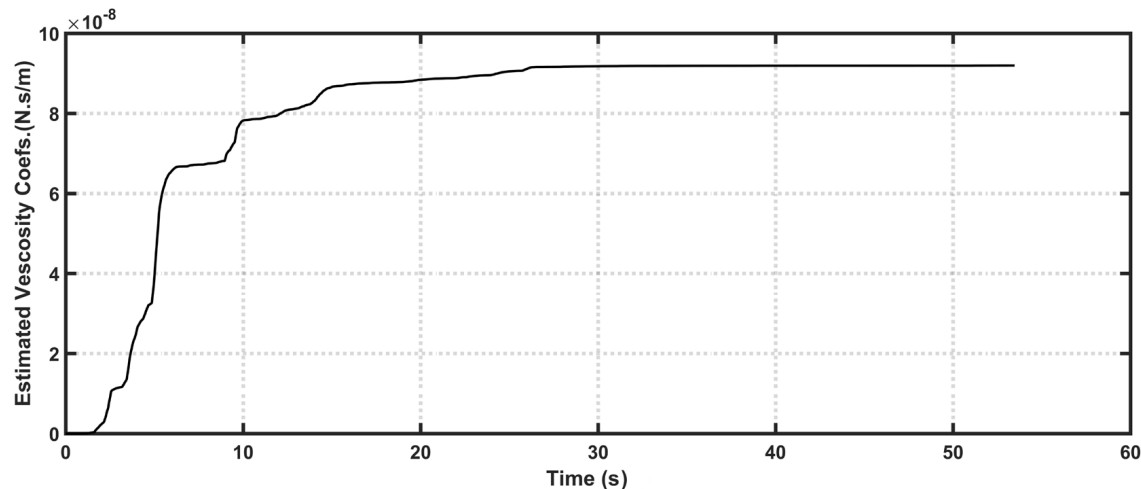

**Figure 10.** Behavior of estimates of the building viscosity coefficient.

## 5. Conclusions

This study presented the design of an adaptive control scheme based on synergetic theory to reduce the effect of a building structure's vibration due to earthquake. A Lyapunov-based stability analysis was conducted to develop the control algorithm so as to guarantee the stability-controlled building system. In addition, a comparison in performance was made between ASC and SC. The numerical simulations showed that better vibration suppression was obtained using ASC as compared to its counterpart. Moreover, the design of the adaptive synergetic controller resulted in a control law and adaptive laws that led to bounded estimates of the building viscosity damping and stiffness coefficients.

This study can be extended to apply the proposed controller to buildings of multiple stories, instead of one-story buildings, or reconsidered for RC heritage buildings. In addition, techniques based on frequency analysis can be conducted either for identification

or for tuning purposes [41–43]. Moreover, the proposed controller can be implemented in a real environment in an actual building or a prototype one. Further, filtered-based synergetic control design of a building system could be conducted to improve the noise rejection capability of the proposed controller [44–46].

**Author Contributions:** Methodology, A.Q.A.-D.; software, A.J.H.; validation, M.E.S.; formal analysis, A.Q.A.-D.; investigation, A.J.H.; resources, Z.T.A.; writing—original draft preparation, M.E.S.; writing—review and editing, Z.T.A.; visualization, A.J.H.; supervision, Z.T.A.; funding acquisition, A.Q.A.-D.; data curation, M.E.S.; conceptualization, A.J.H. All authors have read and agreed to the published version of the manuscript.

**Funding:** This research received no external funding.

**Institutional Review Board Statement:** Not applicable.

**Informed Consent Statement:** Not applicable.

**Data Availability Statement:** Not applicable.

**Conflicts of Interest:** The authors declare no conflict of interest.

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
