# Peer review of "Earthquake Hazard Mitigation for Uncertain Building Systems Based on Adaptive Synergetic Control"

_asi, doi:10.3390/asi6020034_

Round 1

Reviewer 1 Report

This paper presents the design of an adaptive control scheme based on synergetic control theory for suppressing the vibration of building structure due to earthquakes.

The paper investigates an interesting topic.

The structure of the paper is quite clear. However, this reviewer raised some comments that must be addressed before considering this paper for publication.

1.       At the end of the introduction section, an outline of the paper would help the reader and increase the readability of your work.

2.      It is not clear if this system can be calibrated, for instance, with response spectra in order to avoid the activation of resonance modes.

3.      Text must be polished.

4.      This paper is quite theoretical however, in the conclusion section, the authors can make a contribution about the possible implementation of such a system in order to mitigate the risk od a variety of rulings, such as RC heritage building etc.

5.      I recommend using references such as:

https://doi.org/10.1002/pse.208

https://doi.org/10.1615/IntJMultCompEng.2021040212

https://doi.org/10.1016/j.engstruct.2021.112524

Author Response

Dear Sir,

The authors express their gratitude for the reviewer’s great efforts and expert comments for improving our manuscript. Kindly find below the step-by-step answers to your valuable comments:

Comment #1: At the end of the introduction section, an outline of the paper would help the reader and increase the readability of your work.

Response to Comment #1: The outline of the paper has been added at the end of Introduction section. Thank you your kind reminder.

Comment #2: It is not clear if this system can be calibrated, for instance, with response spectra in order to avoid the activation of resonance modes.

Response to Comment #2: The spectrum analysis of vibrated structure subjected to resonant forces due to earthquake is an important field to be indulged for this application. However, this analysis is beyond the scope of this study and the adaptive control based on new control methodology (synergetic control) is proposed instead for vibrated building subjected to uncertainties in their parameters. However, the frequency analysis for vibrated structure and identification based on this aspect is an interesting topic to be studied in the future works. Therefore, this has been highlighted as suggestion for future wok in the conclusion part.  

Comment #3: Text must be polished.

Response to Comment #3: The manuscript has been proofread and it has been polished. Thank you.

Comment #4: This paper is quite theoretical however, in the conclusion section, the authors can make a contribution about the possible implementation of such a system in order to mitigate the risk of a variety of rulings, such as RC heritage building etc.

Response to Comment #4: Thank you for this important and practical point. The authors agree with this point, but since we have introduced a new control methodology, the theoretical basis of control design was of our first concern and the adaptive scheme was a challenging problem to be combined with this control scheme. However, there is a prototype of building is being established in control Lab to be as practical test for future work. In addition, a future work has been added at the end of conclusion section to highlight this important point.    

Comment #5: I recommend using references such as:

https://doi.org/10.1002/pse.208 https://doi.org/10.1615/IntJMultCompEng.2021040212,https://doi.org/10.1016/j.engstruct.2021.112524

 Response to Comment #5: The authors thanks the reviewer for these important works. These suggested works have been cited as future extension of this study. Thank you again.

Reviewer 2 Report

This work presents a vibration absorption scheme, using a magneto-rheological damper.

My observations are the following:

1.- It is important to note the contribution concerning other absorption schemes such as those based on sliding regimes and positive feedback of the position (PPF).

Enriquez-Zarate, J.; Abundis-Fong, H.F.; Silva-Navarro, G. Passive vibration control in a building-like structure using a tuned-mass-damper and an autoparametric cantilever beam absorber. In Active and Passive Smart Structures and Integrated Systems 2015; International Society for Optics and Photonics: Washington, DC, USA, 2015; volume 9431

2.- It is also important to point out the advantages over modal type schemes, based on modal parameter estimation techniques such as:

Trujillo-Franco, L.G.; Silva-Navarro, G.; Beltran-Carbajal, F.; Campos-Mercado, E.; Abundis-Fong, H.F. On-Line Modal Parameter Identification Applied to Linear and Nonlinear Vibration Absorbers. Actuators 2020, 9, 119.

Author Response

Dear Sir,

The authors express their gratitude for the reviewer’s great efforts and expert comments for improving our manuscript. Kindly find below the step-by-step answers to your valuable comments:

Comment #1: It is important to note the contribution concerning other absorption schemes such as those based on sliding regimes and positive feedback of the position (PPF). Enriquez-Zarate, J.; Abundis-Fong, H.F.; Silva-Navarro, G. Passive vibration control in a building-like structure using a tuned-mass-damper and an auto parametric cantilever beam absorber. In Active and Passive Smart Structures and Integrated Systems 2015; International Society for Optics and Photonics: Washington, DC, USA, 2015; volume 9431

Response to Comment #1: The literature review has been enriched with this important study as highlighted by red color. Thank you.

Comment #2: It is also important to point out the advantages over modal type schemes, based on modal parameter estimation techniques such as:  Trujillo-Franco, L.G.; Silva-Navarro, G.; Beltran-Carbajal, F.; Campos-Mercado, E.; Abundis-Fong, H.F. On-Line Modal Parameter Identification Applied to Linear and Nonlinear Vibration Absorbers. Actuators 2020, 9, 119.

Response to Comment #2: The parameter estimation technique is very important in vibration suppression and it is instructive in enriching the state of art. Therefore, this important work has been added to the related works. Thank you.

Reviewer 3 Report

1. The introduction section should be revised. It is not an introduction of the publihsed work.  Comments on the published work should be added.

2. The authors should explicitly explaine the novelty of their work.

3.  Since the proposed controller can be applied to building of multiple stories instead of one-story building, MDOF system should be takend as an example.

Author Response

Dear Sir,

The authors express their gratitude for the reviewer’s great efforts and expert comments for improving our manuscript. Kindly find below the step-by-step answers to your valuable comments:

Comment #1: The introduction section should be revised. It is not an introduction of the published work.  Comments on the published work should be added

Response to Comment #1: The introduction has been revised and proofread. In addition, the motivation behind using the proposed synergetic control in this application has been highlighted and focused in red color immediately after the literature review. Many thank for this expert comment.  

Comment #2: The authors should explicitly explained the novelty of their work.

Response to Comment #2: Thank you for this important point. The novelty of this study has been mentioned and indicated in the introduction part. Thank you again for your kind reminder.

Comment #3: Since the proposed controller can be applied to building of multiple stories instead of one-story building, MDOF system should be takend as an example.

Response to Comment #3: The authors agree with this point, but the development of adaptive control algorithm based on synergetic control for multiple DOF is a challenging problem and it takes a time. This is due of the complexity of model and the difficulty of developing adaptive and control laws unless some assumptions will be made. Actually, the authors have started and they will proceed with control design based on this complex model and a generalized control design is intended. However, this point will be considered in the next study and this has been highlighted as future work at the end of conclusion section. Thank you for your kind consideration. 

Round 2

Reviewer 1 Report

The paper can be accepted in the present version.

I would just recommend polishing the text.

Reviewer 2 Report

I am satisfied with the authors responses and improvements.

Reviewer 3 Report

The  manuscript can be accepted in present form.